# The Role of Diffusion Weighted MR Imaging in the Diagnosis of Tendon Injuries of the Ankle and Foot

**DOI:** 10.3390/medicina58020321

**Published:** 2022-02-20

**Authors:** Hasan Aydın, Volkan Kızılgöz, Önder Ersan, Baki Hekimoğlu

**Affiliations:** 1Radiology Department, SBU Ankara Oncology Education and Research Hospital, Ankara 06110, Turkey; 2Radiology Department, Erzincan Binali Yıldırım University, Erzinkan 24000, Turkey; volkankizilgoz@gmail.com; 3Orthopedics and Traumatology Department, SBU Dışkapı Yıldırım Beyazıt Education and Research Hospital, Ankara 06110, Turkey; onderersan@gmail.com; 4Radiology Department, SBU Dışkapı Yıldırım Beyazıt Education and Research Hospital, Ankara 06110, Turkey; bakihekim@gmail.com

**Keywords:** diffusion weighted MRI, diffusion MRI, tendon injuries, MRI, ankle injuries

## Abstract

*Background and objectives:* Our aim is to determine the diagnostic performance and utility of Diffusion Weighted MR Imaging (DWI) against the routine Magnetic Resonance Imaging (MRI) for the evaluation of patients with tendon injuries of the ankle and foot. *Materials and Method:* After institutional review board approval and informed consent taken from all the patients, ankle and foot MR imaging and DWI-Apparent Diffusion Coefficient (ADC) mapping were performed on the 81 injured tendons of 50 patients. All tendon injuries were named as Rupture (R), Partial tear (PT), and Tenosynovitis (T). Diagnostic interpretation was based on the MRI-DWI and ADC mapping, verified by either open surgery, diagnostic arthroscopy, or conservative procedures-splint application. Statistical analysis of this research was assessed by Fischer’s exact test, variance analysis test between dependent groups, Receiver Operating Characteristics (ROC) curve, and Pearson chi square statistics. *Results:* MRI depicted all tendon injuries with 70% sensitivity and 100% specificity, and showed a significant statistical relationship to surgical and arthroscopic references with high agreement (*p* < 0.05, k: 0.609). DWI had 100% sensitivity and 83–90% specificity for the visualization of tendon injuries with certain agreement and a significant statistical relationship to the gold standard (*p* < 0.05, k: 0.890–0.899). For all those injured tendons, DWI had 100% sensitivity for the diagnosis of R, and 92–97% sensitivity corresponding to PT and T over routine ankle MR imaging. The specificity of DWI to MRI ranged from 75 to 44% for all the injured tendons. DWI had significant statistical superiority over MRI for the visualization of R, PT, and T of all tendons included in this research (*p* < 0.05). *Conclusions:* DWI is a good imaging modality for the visualization of ankles with tendon injuries, possibly further improving the sensitivity of the classical ankle and foot MRI, and supplying more beneficial and diagnostic information than routine MR imaging on the basis of R, PT, and T of tendons at the ankle and foot.

## 1. Introduction

Clinical examination and conventional radiography are the first steps in evaluating whether a tendon or a ligament has been injured in the foot and ankle, but even those are not enough nor helpful in highlighting these injuries, especially when those injuries are not associated with bone avulsions and large intra-articular effusions [1,2]. For assessing pathological conditions of foot and ankle, Magnetic Resonance Imaging (MRI) has a substantially increasing role with the advancement of new software techniques and new sequences [1,3]. Potential advantages of MRI include its multiplanar imaging capacity, its high sensitivity in distinguishing normal and pathological tissues, and its capability of simultaneously evaluating tendons, ligaments, nerve, and fascial injuries of the ankle due to its high soft-tissue contrast resolution [1,3,4].

Lateral ankle ligament complex injuries (anterior talofibular ligament (ATFL), calcaneofibular ligament (CFL), and posterior talofibular ligament (PTFL)) are present in up to 85% of ankle sprains, making them the most frequently injured ligaments secondary to strong ankle inversion movement during lateral ankle sprains [5]. PTFL injury is infrequent, on the other hand, ATFL is considered as the principal injured ligament followed by the CFL [5,6]. The medial ankle complex is composed of the tibialis posterior tendon, spring ligament (SL), deltoid ligament, and tibiotalar ligament; SL is considered as the strongest medial ligament since this ligament supports compression forces [5,6,7]. MRI is a precise, reliable, and valid tool to visualize and diagnose those ankle injuries [5,6,7,8].

MRI can be successfully utilized for the diagnosis of ankle instability, impingement, and injuries of tendons and ligaments; useful for the assessment of a wide range of soft-tissue and osseous disorders of the ankle, thereby helping the surgeon to plan an appropriate treatment; however, ankle instability and ankle impingements are diagnosed mostly based on symptoms and physical examination [4,8,9,10,11], but in our view, the exact diagnosis of tendon injuries, especially the partial tear (PT) and tenosynovitis (T), may require some other imaging modalities. This is mostly due to the need to increase the sensitivity and specificity of MRI, which ranges between 60 and 80% [1,2,9,10,11,12,13]; one can easily misdiagnose a tendon injury in every three ankles and tendon sprains of both feet by these results.

Rehabilitative Ultrasound Imaging (RUSI) has been used to measure the Cross-Sectional Area (CSA), thickness, and connective tissue of various muscles associated with musculoskeletal conditions that affect physical therapy and conservative evaluation [14]. CSA and thickness of plantar muscles (flexor hallucis brevis; flexor digitorum brevis; abductor hallucis; flexor digitorum longus; flexor hallucis longus; tibialis anterior; and peroneus longus and peroneus brevis) and fascia can be used to explain how injuries may be related to alterations in foot function and clinical pathology [14,15]. Atrophy, tendon rupture, and delayed reaction times of the peroneus muscles have been related to ankle sprains [14,16]. Changes in posture and hip abductor muscle strength may be generated after inversion ankle sprain. Therefore, the consequences of ankle injury may affect proximal structures of the lower limb; in the presence of functional ankle instability, activation patterns of the lower limb proximal muscles may be altered [17].

With the advancing technology, new MRI sequences have been proposed to improve the quality of routine MRI protocol and have been applied to increase the sensitivity and specificity for the diagnosis of ankle injuries, and at the same time, to overcome the misdiagnosis of tendon injuries, especially due to the failure of MRI [3,12,13].

Such a new sequence: Diffusion Weighted Imaging (DWI) and Apparent Diffusion Coefficient (ADC) mapping will be analyzed in this study, our hypotheses is that DWI aids in the diagnosis of tendon injuries of the foot and ankle; its use beneath the routine ankle MRI can easily improve and strengthen the diagnostic performance of MRI. In our view, the scientific contribution of this paper to the literature is that DWI can supply adequate information in the diagnosis of ankle sprains and can collaborate with the routine MRI, which can support the early diagnosis by orthopedicians, resulting in a reliable therapeutic approach for the benefit of ankle trauma patients. Our aim is to define the utility and capacity of DWI-ADC mapping for the diagnosis of tendon injuries in the ankle and foot, with regard to routine ankle MR imaging.

## 2. Methods

### 2.1. Study Design

Between November 2012 and August 2020, 50 patients (16 males and 34 females, ages ranging between 20 and 74 years, mean: 50 years) with ankle and foot sprains were prospectively evaluated.

### 2.2. Participants

A total of 23 left, and 27 right, ankles were included in this research. All patients underwent a detailed history and physical examination by an experienced orthopedic surgeon and his four colleagues, then referred to us for an MRI of the ankle. All patients had precise pain and ankle instability mostly due to a trauma at walking and sports and/or due to jogging injuries and accidents. Besides tendon injuries, there were also bone bruises, fractures and microfractures of bones, ligament tears, synovitis, synovial thickening, and bursitis in some patients, but none of these patients were excluded. None of the patients had a history of rheumatoid arthritis or any other sero-negative arthritis. Written informed consent was obtained from all patients prior to imaging, and the institutional review board approved the research 

### 2.3. Test Methods

Achilles tendon (A), tibialis posterior tendon (TP), peroneal tendons (PT), tibialis anterior tendon (TA), extensor digitorum longus tendon (EDL), extensor hallucis longus tendon (EHL), flexor hallucis longus tendon (FHL), and flexor digitorum longus tendon (FDL) injuries were all analyzed in this research. Peroneus longus and brevis were taken as one peroneal tendon, injury of either tendons or both, were taken as PT injury with regard to the comments and referral of the orthopedicians. A total of 81 tendon injuries of ankle and foot were evaluated in this report; 20 patients in this report had at least one tendon injury.

MRI and DWI studies were performed on a 1.5 T MR scanner (Philips Achieva, Philips Medical systems, Eindhoven, The Netherlands and 32 channel GE Signa GE, Milwaukee, Brookfield, WI, USA) by using an 8-channel standard transmit receive extremity coils with a linear configuration. Patients were laid in a supine position, with the ankle placed in a mild plantar flexion. Our MRI protocol: T1 weighted imaging (T1W) spin-echo (SE) sagittal/coronal planes (400–500/15–25 TR/TE), slice thickness 3 mm with 0.5 mm gap interface, 12–16 cm FOV, 192x256 matrix size, duration of scan: 2.15/1.53 min. T2 weighted imaging (T2W) turbo SE with fat-saturation (FS) sagittal and axial planes (3650/75-TR/TE), echo train length 8–12, slice thickness 4 mm with 0.4 mm gap interface, 14–17 cm FOV, 256 × 256 matrix size, imaging time: 2.40 min for each plane. PD-FS-coronal plane (2650/30-TR/TE), echo train length 6–10, slice thickness 4 mm with 0.4 mm gap interface, 15–17 cm FOV, 256 × 256 matrix size, and imaging time: 1.55 min. DWI and ADC mapping were performed by echo-planar spin echo imaging with 3D acquisitions, mostly by axial images as transverse planes were best to evaluate structural abnormalities of tendons with b values 400–600-800 s/mm^2^ (3700/65 msec-TR/TE), 15 cm FOV, 3 cm slice thickness, time of scan for each plane was 2.30 min. ADC mapping was acquired from one of these b-values with higher image quality. All MRI acquisitions were performed in the radiology departments of hospitals.

Two musculoskeletal radiologists analyzed the MRI images of ankle and foot together; a senior radiologist with 15 years’ experience and a special radiologist with 10 years’ experience. These MRI sequences and images were evaluated after the analysis of routine radiographs and there was no disagreement in the interpretation of the MR images. Two to three days later, both radiologists reviewed DWI-ADC mapping studies on separate work-stations independently and any disagreement was resolved by means of consensus, reviewers were blinded to patients’ data, clinical history, and surgical results, but were aware of the outcomes of prior MR sequences regarding the injured tendons. Both readers evaluated all DWI data in a random patient case order, quite different than the order of the MR images in order to overcome the potential bias for the interpretation of both sequences and not to cause any disadvantages for the real potential of DWI to diagnose tendon injuries. Senior radiologist with 15 years’ experience and senior orthopedician together determined whether the ankle MRI met the inclusion criteria of the patients belonging to this study. This senior radiologist also conducted each session of the evaluation trials. There were 2 sessions with one trial for each, both for MRI and for DWI evaluations; in total, 4 sessions with 2 trials were applied for both observers.

Inter-rater reliability (also called inter-rater agreement, inter-rater concordance, inter-observer reliability, etc.) is the degree of agreement among independent observers who rate, code, or assess the same phenomenon. In contrast, intra-rater reliability is a score of the consistency in ratings given by the same person across multiple instances. In other words, reliability of a measurement refers to the consistency of the data when the same trait is measured by the same observer (intra-rater reliability) or by different observer’s (inter-rater reliability) with the same measurement device [18,19,20].

It is very important to establish inter- and intra-observer reliability when conducting observational research, it refers to the extent to which two or more observers are observing and recording behavior in the same way [20].

In order to calculate inter-rater reliability, each session was carried out by the observers’ on the same day of the MRI and DWI evaluations. Achievement of the equal conditions for both observers during the experimental evaluation was assessed by offering them almost the same workstations with similar properties independently and if present, by solving the problems with consensus. As a 2–3 day interval was taken for DWI analysis after the interpretation of the MR images, this period was used to assess intra-rater reliability on the same day for both observers due to their high diagnostic performance rate on that day in the evaluation of all images. In other related studies in the literature in which those procedures were applied, time interval ranged from 1 to 2 h/week for 1–2 months in order to ensure proficiency with the MRI and/or USI [21,22]. We did not need any other traditional methods to assess inter- and intra-observer reliability as we thought that these used methods were reliable and eligible for our reported measures.

### 2.4. Analysis of Data Set

All tendon injuries were named as Rupture or Complete tear (R), PT, and T. Thickening of tendons, >6 mm for A and >3 mm for other tendons with convex bulging of their anterior margins, without any intratendinous high signal on T2W and DWI and without any disruption/retraction or discontinuity and fluid of tendon sheath with more than 5 mm width, was recorded as **T**; fluid of tendon sheaths with less than 5 mm width was regarded as simple sheath effusion and was not involved in this research [3,5,10,11,12]. Thickening of tendons with clefts and defects, partially extending into the substance, irregular areas of high signal intensity larger than 3 mm on T2W and DWI, or slight indistinctness and waviness without retraction were presented as **PT** [3,5,8,12]. Complete disruption and retraction with high signal intensity on T2W-DWI between torn edges, absence of tendons, and division of tendons into distinct slips with or without heterogeneity were defined as **R** [3,9,12,13].

In the Scanning Electron Microscopy (SEM), most of the collagen fibrillar bundles ran straight and parallel to the long axis of the stretched tendons: flattened and partially crimped bundles were recognizable in only a few segments, in terms of PT of tendons; completely straightened collagen fibrils with residual knots corresponding to ‘fibrillar crimps’ with flattened and totally stretched crimps, in terms of R of tendons; and collagen fibril bundles arranged in a crimp pattern with a sharp top angle with fluid accumulation through the tendon sheath, in terms of T of tendons [23].

General information about the T, PT, and R, defined in the above paragraph, were considered to be the reference values to test the diagnostic accuracy of imaging approaches. Only injured tendons were included in the research; however, both observers defined 1 T case as normal/healthy tendon. All measurements of DWI were performed from the longest diameter. Open surgery was performed for 9 ruptured tendons, diagnostic arthroscopy and splint application were performed for 41 tendons with PT. Diagnosis and treatment of T of 31 tendons were based on physical examination, conservative therapy, and follow-up splint or bandage application. Conservative treatment included bed rest, non-steroidal anti-inflammatory drugs, and stretching. Corticosteroid injections were rarely used.

All arthroscopic procedures were performed by a 15-year-experienced orthopedic surgeon and four of his team-mates (two specialized orthopedicians and two trainees). Through the 81 injured tendons, 50 tendons with R and PT used the surgical approach as the reference standard (62.5%). None of the patients in R and PT groups refused the arthroscopy. Six patients in T group refused the splint application and all other T group patients were treated conservatively; those six patients were not excluded from the study.

### 2.5. Statistical Analysis

Statistical analysis of this research was performed by SPSS 25 written form (SPSS Inc., Chicago, IL, USA). This analysis compared the independent DWI and MRI results of both readers and the common observers list with consensus, intra and interobserver variability for all injuries were calculated by variance analysis test with Kappa (k) values, Cohen’s κ model weighted kappa, regarding the consensus of DWI. k values more than 0.80 indicated perfect agreement, k from 0.6 to 0.8 presented high agreement, k values between 0.4 and 0.6 indicated moderate agreement, and k values of 0.2 to 0.4 presented fair agreement [9,10]. Correlation of surgical findings to DWI and MRI results of both readers were analyzed by Fisher’s exact *t*-test with *p* value, sensitivity, and the specificity ratios of both modalities—positive/negative predictive values were also additionally calculated by this test, with regard to arthroscopy as the reference. Comparison of DWI and MRI results of both readers against surgical results were performed by Pearson chi-square test. For all tests, *p* < 0.05 was considered to indicate a statistically significant difference. Pearson chi-square test was the gold standard test for the evaluation of DWI and MRI results of both readers.

As the inter/intra-observer variabilities were tested and had statistically valuable importance to this article, the confidence level of each rater was considered to be a new diagnostic modality for the tendon injuries. Therefore, ROC curves were fitted to reveal the sensitivity and specificity of DWI and MRI more precisely with regard to those confidence levels and diagnostic performances of both observers, which were estimated by using area under this curve (AUC). Sensitivity and specificity ratios were also plotted under this area by using ROC curves.

## 3. Results

A total of 9 ruptured tendons (1 EDL, 1 FHL, 1 PT, and 6 A), 41 partially torn tendons (2 EHL, 4 EDL, 3 FDL, 16 FHL, 15 PT, and 1 A), and 31 tendons with **T** (1 EHL, 8 FDL and FHL, 3 PT, and 11 TP) were involved in this research. Eighty-one tendon injuries and sprains were analyzed statistically. For **R** of EDL, FHL, PT, and A; **T** of EHL, FDL, FHL, PT, and TP; and for the **PT** of all tendons except TA and TP, both observer’s independent results were all similar with high agreement, had a significant statistical relationship with each other and with the common DWI list, and were correlated with each other and with the common DWI list (k: 0.709, *p* < 0.05). This was also proved statistically for the injured PT and FHL tendons with regard to the independent results of each reader, and correlated to the common DWI findings with moderate to high agreement (*p* < 0.05, k: 0.415–0.714) [24,25]. Six ruptured A tendons were also recognized by DWI with perfect agreement without any statistical differences from the surgery (*p* > 0.05, k: 1.00) (Table 1). As there was only one PT of A, we had no statistical correlation, but DWI had 100% success for the depiction of this injury.

For **R**, **T**, and **PT** of EDL, EHL, FDL, TP, and TA, there was no statistical correlation for each reader’s results to the common DWI list (*p* > 0.05). For **R**, **T**, and **PT** of FHL and PT, findings of both readers had a significant statistical relationship correlated with the results to each other (*p* < 0.05, k: 0.514–0.744, moderate to high agreement). Results of both readers for other injured tendons did not present any statistical relationship to each other with high to perfect agreement (*p* > 0.05, k: 0.547–1.2) (Table 2a,b, Figure 1a,b and Figure 2).

MRI depicted all tendon injuries with 70% sensitivity and 100% specificity, and had significant statistical correlation to the surgical arthroscopic references with high agreement (*p* < 0.05, k: 0.609). MRI sensitivity ranged from 50 to 89% for all tendon injuries (50% for EHL and FHL, 89% for FDL, 83% for A, 75% for EDL, 64% for PT, and 90% for TP) with almost 100% specificity for the R, T, and PT of all tendons. In this research, MRI showed highest specificity for the diagnosis of all injured tendons. Regarding the R of nine tendons (1 PT, EDL, and FHL, and 6 A), there was no statistical correlation for each of the tendons separately, and DWI had 100% sensitivity and specificity for the diagnosis of all ruptured tendons without any statistical differences against the surgical approaches with perfect agreement (*p* > 0.05, k: 1.00). DWI had 100% sensitivity and 90% specificity for the visualization of 41 tendons with PT, and 100% sensitivity and 83% specificity for the evaluation of 31 tendons with T, by certain agreement and with a significant statistical relationship over the gold standard, arthroscopy, and physical examination (*p* < 0.05, k: 0.890–0.899) [24,25].

For 16 PT of FHL tendons and 15 PT of PT, DWI had 100% sensitivity for both tendon injuries, 88% specificity for FHL, and 100% for PT with high to perfect agreement and a significant statistical relationship to the reference (*p* < 0.05, k: 0.875–1.00). Except for the T of FHL tendons (100% sensitivity and 67% specificity), all other PT and T of tendons were diagnosed by DWI with perfect agreement and 100% sensitivity/specificity, without any statistical approval over the gold standard or diagnostic arthroscopy (*p* > 0.05, k: 1.00) [24,25] (Table 3a–c; Figure 3a–c).

R, PT, and T are the main sections of diseased tendons and probable diagnostic rationales for the clinicians who are faced towards these entities with only positive test results.

When we correlate the results of DWI over MRI for the depiction of tendon injuries, the PT and T of all tendons were diagnosed by DWI with 92–97% sensitivity and 55–44% specificity; they had significant statistical correlation against MRI with fair to moderate agreement (*p* < 0.05, k: 0.362–0.459) (Figure 4), and they predicted significant statistical superiority over routine MRI with higher sensitivity and less specificity; thus, our hypotheses, using DWI in addition to the routine MRI protocol for the exact diagnosis of PT and T of tendons, might be considered reliable and clinically relevant following these statistical results (Figure 5a–d). DWI presented 100% sensitivity and 75% specificity for the diagnosis of ruptured tendons with high agreement and had significant statistical differences from the MRI, and superiority over routine ankle MRI (*p* < 0.05, k: 0.769) was also found. DWI had also statistical superiority over MRI, with fair to moderate agreement for the evaluation of **T** of FDL, and **PT** of FHL and PT (*p* < 0.05, k: 0.284–0.483). Sensitivity of DWI ranged from 50 to 100% and specificity from 12.5 to 100% for the visualization of EDL, EHL, and TP injuries with fair to perfect agreement without any statistical dominancy over the MRI (*p* > 0.05, k: 0.20–1.00) (Figure 6 and Figure 7).

When we evaluate the results of both readers by the ROC curve, R of tendons had an AUC value: 0.875 ± 0.141, with 100% sensitivity and specificity for both readers. Sensitivity remained constant (100%); specificity ranged from 25 to 100% for the AUC values between 0.500 and 1.000. PT of tendons had an AUC value 0.734 + 0.080, and 100% sensitivity and specificity were found for both readers, by AUC value: 0.500, 91.7% sensitivity, and 44.8% specificity were predicted. Sensitivity ranged from 91.7 to 100% and specificity between 44.8 and 100% for both readers with AUC values between 0.500 and 1.000.

AUC value for T of all tendons was 0.699 + 0.116 and from this value to AUC: 1.000, indicating 100% sensitivity and specificity. AUC value: 0.500, presented 95.5% sensitivity and 55.6% specificity for both readers. Sensitivity ranged from 95 to 100%, specificity ranged from 55 to 100% between these AUC values for both readers (Figure 8).

The first reader had one and the second reader had two false negative results for the R of tendons. Both readers had twelve false negatives for PT of tendons. The first reader had nine and the second reader had five false negatives for the T. Both readers had no false positives for ruptured tendons. For PT of tendons, the first reader had six, and the second reader had eight false positive results. For the T, the first reader had seven, and the second reader had three false positive yields.

## 4. Discussion

Conventional ankle MRI has been widely used for the assessment of all ligament and tendon injuries, and it has also used to differentiate tendon abnormalities from other causes of ankle pain, such as occult fractures, bone contusions, osteochondral injury, etc. [10,12,26,27,28]. Thickened tendons with intratendinous high signal intensity on T2W images, discontinuous or torn edges of tendons with contour irregularities and wavy or fascicular appearance, and high fluid accumulation in tendon sheaths were the common MRI findings of injured tendons of ankle and foot [26,28,29,30,31,32,33].

There were numerous reports in the literature regarding the ankle MRI. Most of the presented reports were about A tendon injuries: Marcus et al. [4] declared 93% sensitivity with MRI for seven A injuries; Karjalainen et al. [26] studied overuse injuries of 118 A tendons and claimed 94% sensitivity for A injuries, and 95% sensitivity for T of A tendons with MRI. Kuwada et al. [1], who studied 28 patients with A, TP, PT, and posterior tibiotalar ligament (PTT) injuries, presented 57% sensitivity for PT ruptures, 73% for PTT, and 94% for A and TP tears. Rosenberg et al. [28] studied 27 patients with longitudinal PT tears, presented 71% sensitivity for peroneus brevis tears with MRI. In our current study, sensitivity of MRI for the detection of tendon injuries of ankle and foot, ranged from 50 to 90%; highest in TP and FDL, lowest in EHL and FHL, 64% sensitivity for PT, and 83 % sensitivity for A; specificity remained 100% for all injured tendons. Our MRI results were more diagnostic than the outcomes of Kuwada et al. [1] for PT tears, but our results for A tendon injuries with MRI were less efficient than Marcus et al. (1989) [4] and Karjalainen et al. [26] (83%/93–94%). Rosenberg et al. [28] also presented higher MRI sensitivity for Peroneus brevis tears than ours (64%/71%).

DWI allows quantitative measurement of the motion of water molecules into the lesions and normal tissues, net diffusion of water molecules is referred as the ADC, and it combines the effects of capillary perfusion and water diffusion in the extracellular space [34,35,36,37]. DWI has been widely used for the detection of perfusion deficiencies in the brain, resulting in cerebral ischemia, but over the last few years, extracranial applications of DWI have been performed for the evaluation of abdominal organ pathologies, especially for the differentiation between benign and malignant lesions of the liver and kidneys, head and neck disorders, and assessment of lymphadenopathies throughout the whole body [36,37,38,39,40,41,42].

Achilles tendon exploration by diffusion tensor imaging, post-operative Achilles tendon examinations by MRI-US, and rotator cuff tears imaging by DWI were relevant in the literature, but, for our purposes, this report was the first in the literature concerning the efficacy of DWI in the diagnosis of tendon injuries in the ankle and foot [39,41,43,44,45].

We thought that DWI theoretically might supplement relevant data beneath the routine MRI, via presenting the motion and diffusion of water molecules through the diseased tendons, showing the accumulated effusion within the tendon sheath plus peritendinous areas, and supplying a good-contrast between the injured tendons and normal surrounding soft tissue. In our view, these contrast differences in the imaging approaches, revealing the diseased tendons, probably supply the most important data for clinical practices, and this was the crucial point of this research. MRI presented 83%, DWI had 100% sensitivity for the evaluation of A injuries, and both imaging modalities had 100% specificity for this injury. Nevertheless, DWI was more sensitive than MRI for the depiction of A rupture without any statistical superiority, so surgeons should not simply use DWI for clinical diagnosis of A tendon disorders; however, with respect to whole ruptured tendons, DWI had significant statistical superiority over MRI in this report.

DWI predicted superiority to ankle MRI for the evaluation of R, PT, and T of all tendons with more than 90% sensitivity and about 50% specificity over MRI. This was mostly due to the visualization of **R** of A, **T** of FDL, **PT** of FHL, and PT 80–100% sensitivity for PT of FHL, and PT with 55–64% specificity and 88% sensitivity for the T of FDL, as observed by DWI. These were quite satisfactory results and, to our belief, could easily influence the results of a routine MRI, and might aid in the treatment planning of clinicians and surgeons. This research also demonstrated that DWI was more valuable than routine MRI for the depiction of ruptured tendons. With regard to arthroscopy, DWI had 100% sensitivity for all injured tendons and presented 67–100% specificity for the diagnosis of them.

MRI regarded 70% sensitivity (50 to 90% range) and 100% specificity for tendon injuries, significantly less sensitive than DWI; thus, we can easily declare that routine MRI protocol of ankle and foot was not enough for the entire diagnosis of tendon injuries, especially for PT and T. Additional imaging techniques such as DWI could aid in the diagnosis, supply more beneficial and relevant data for the depiction of injured tendons, and could also be included in the routine ankle MRI for more appropriate evaluation.

**Limitations of this research were:** T was about 37.5% of all injuries; these were diagnosed and treated conservatively so there was no surgical proof regarding the less relevant results. We did not adopt contrast-enhanced scans in the routine ankle MRI as this might have created some mis- or overdiagnosing tendon injuries. We did not include a control group because it was very difficult to create a strict control group with individuals who had never had an ankle injury. Finally, DWI had lower spatial resolution with low image quality, this might have therefore presented some problems for the visualization and diagnosis of tendon injuries.

In conclusion, DWI with a duration of scan about 5 min, with acquisition in two planes, might further improve the sensitivity, specificity, and accuracy of classical ankle and foot MRI, and might strengthen the clinicians’ and surgeons’ profiles for the treatment options of patients. With regard to the yields of this report, DWI revealed additional diagnostic information about the injured tendons, especially for PT and T.

## Figures and Tables

**Figure 1 medicina-58-00321-f001:**
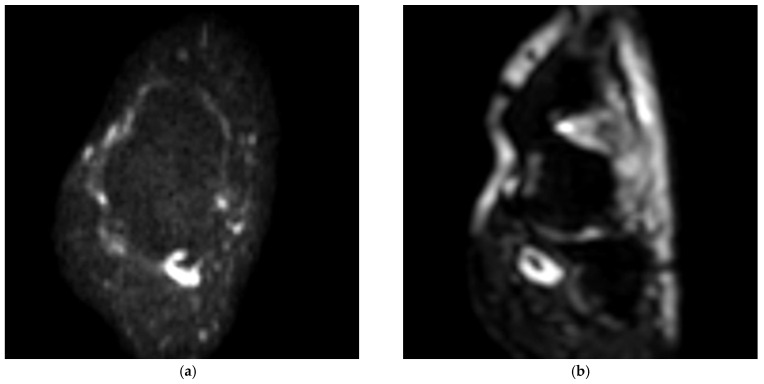
(**a**,**b**) Axial and coronal ADC-DWI images show the FHL tendon partial tear.

**Figure 2 medicina-58-00321-f002:**
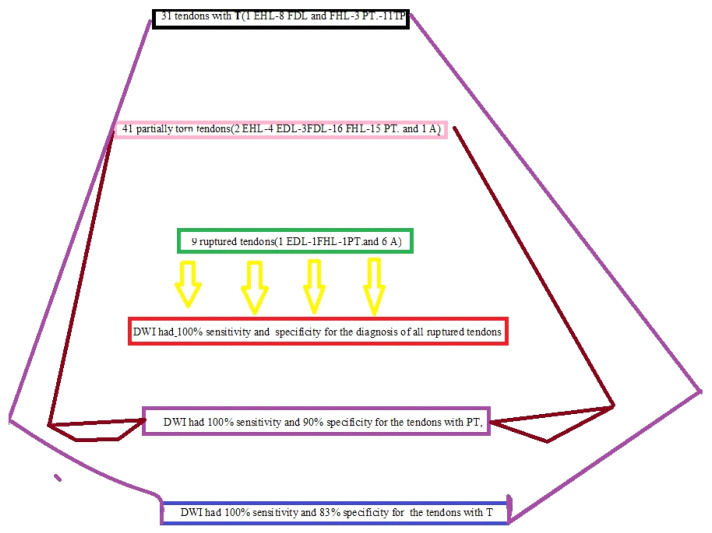
Study Flow Chart for Readers.

**Figure 3 medicina-58-00321-f003:**
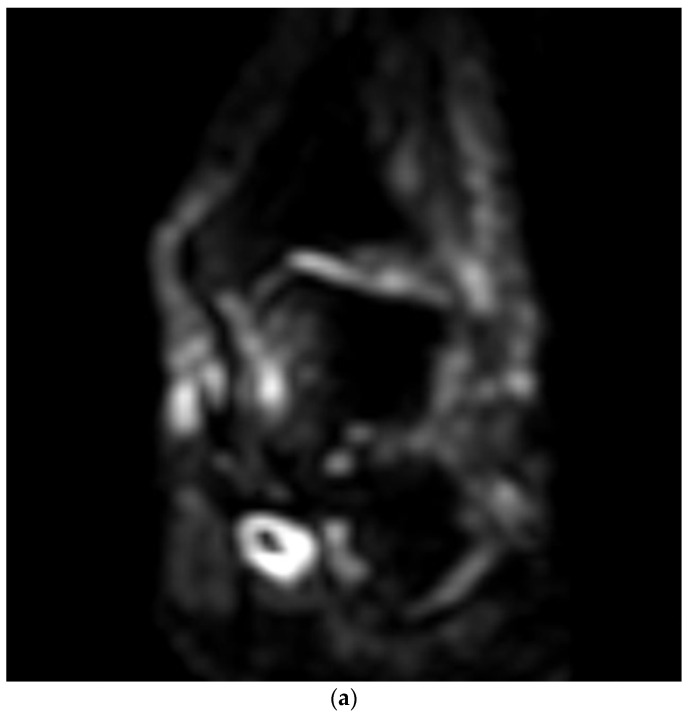
(**a**–**c**) Sagittal T2WI, coronal DWI, axial T2WI images. The images depict an FDL tendon partial tear.

**Figure 4 medicina-58-00321-f004:**
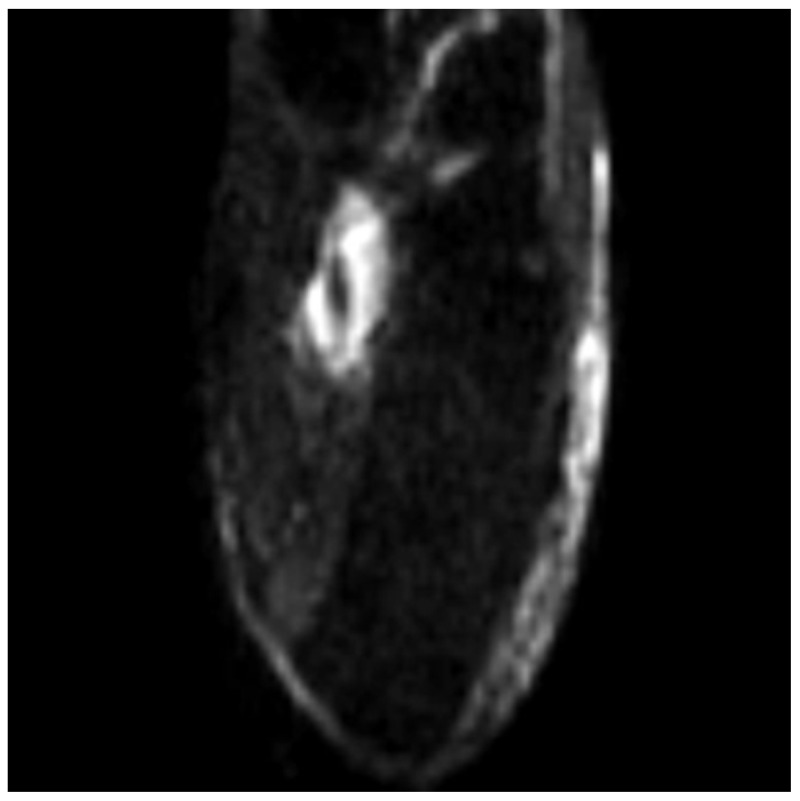
Axial ADC-DWI presents a TP tendon tear.

**Figure 5 medicina-58-00321-f005:**
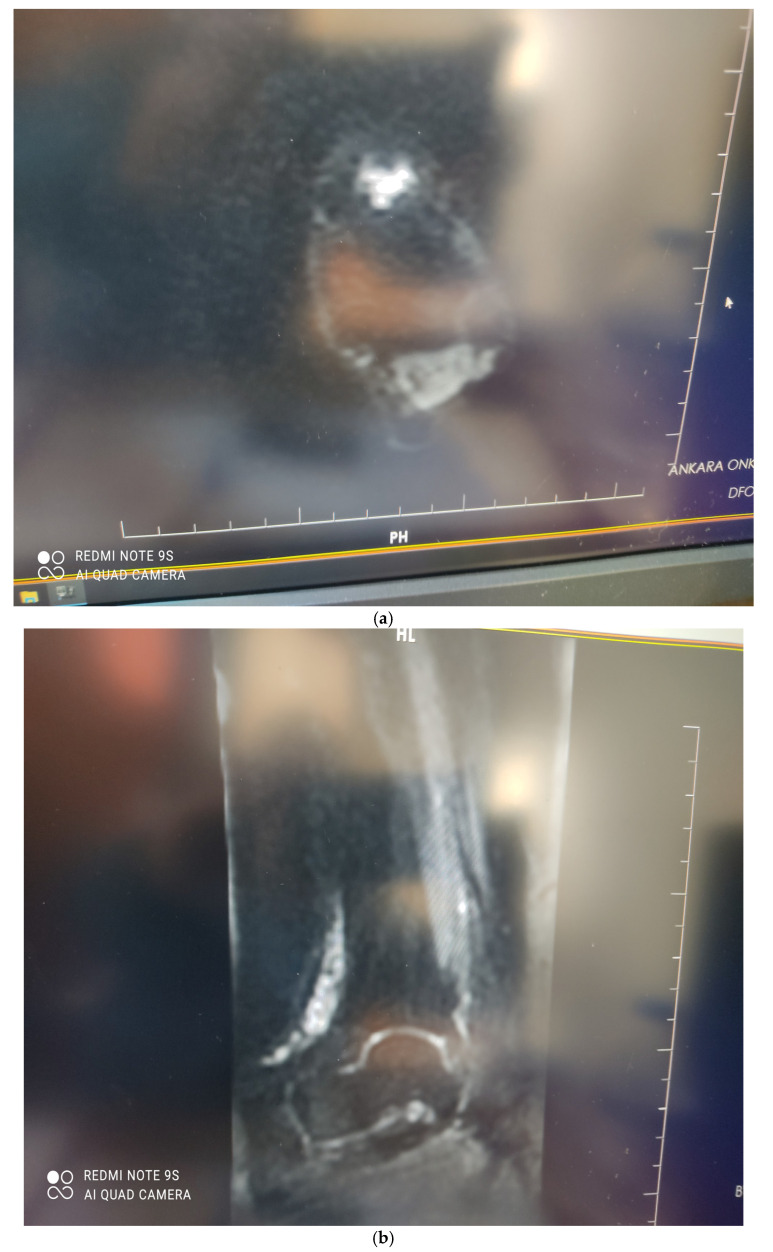
(**a**–**d**) Axial-Sagittal DWI, axial, and Sagittal T2W images show post-sequela cystic TA tenosynovitis.

**Figure 6 medicina-58-00321-f006:**
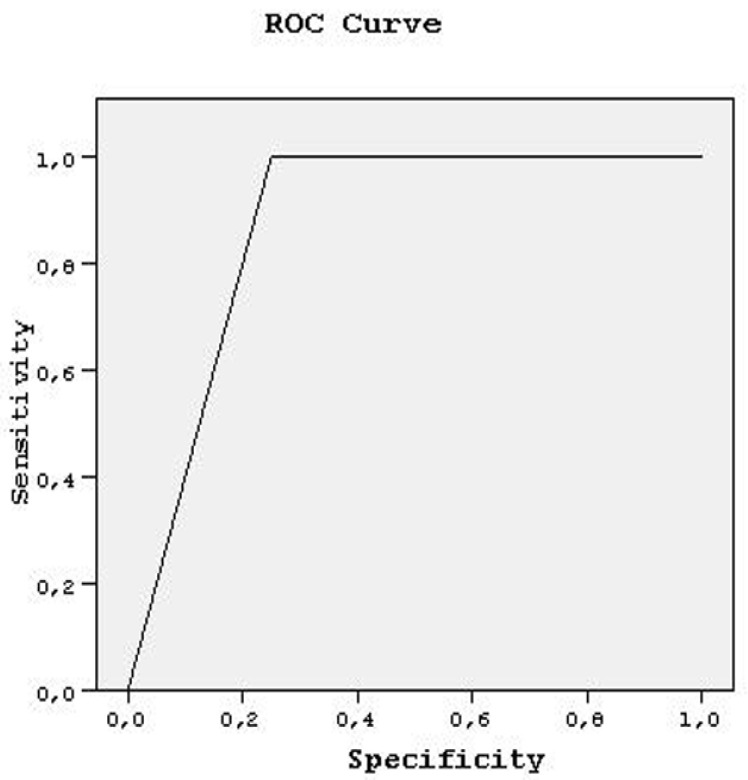
ROC curve analysis, presenting DWI.MRI comparison for ruptured tendons.

**Figure 7 medicina-58-00321-f007:**
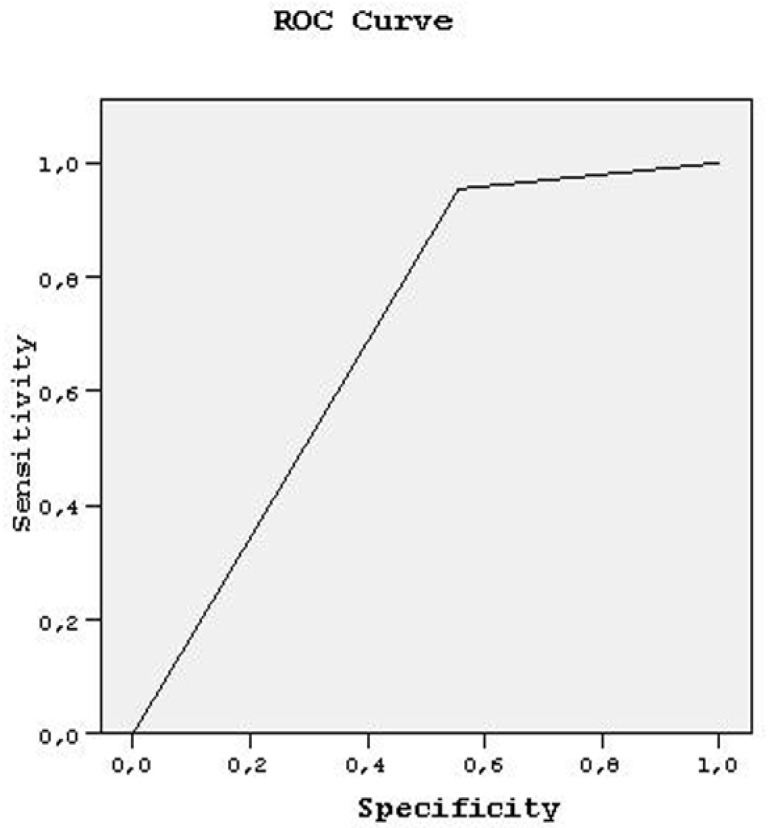
ROC curve analysis, predicting DWI.MRI comparison for tendons with partial tears.

**Figure 8 medicina-58-00321-f008:**
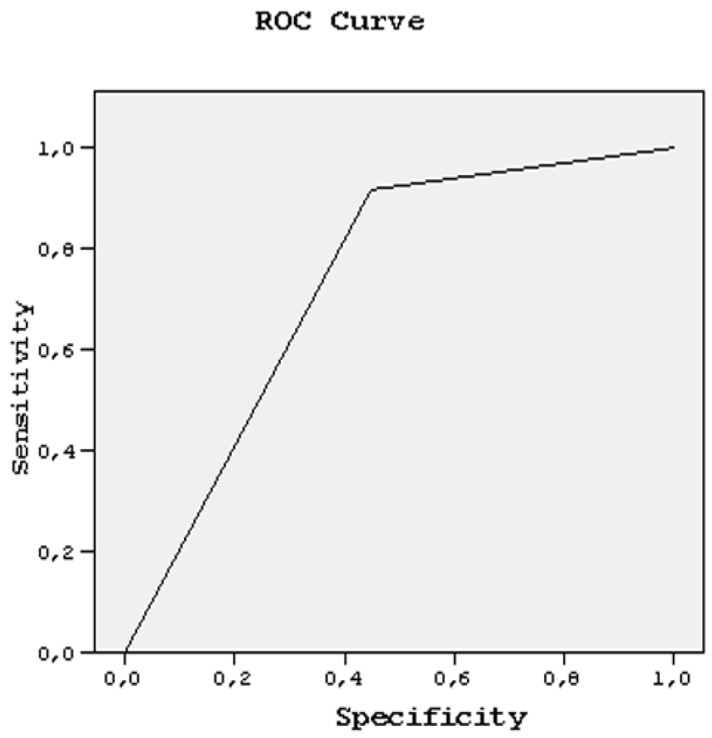
ROC curve analysis, regarding DWI.MRI comparison for tendons with tenosynovitis.

**Table 1 medicina-58-00321-t001:** Statistical relationship between two observers for all tendons.

All Tendons			2. Observer				Total
			Normal	Partial Tear	Rupture	Tenosynovitis	
1. Observer	**Partial-Tear**	*n*	0	29	4	4	37
		%	0.0%	82.9%	5.7%	11.4%	100.0%
	**Rupture**	*n*	0	1	5	2	8
		%	0.0%	20.0%	60.0%	20.0%	100.0%
	**Tenosynovitis**	*n*	1	17	0	18	36
		%	2.8%	47.2%	0.0%	50.0%	100.0%
	**Normal**	*n*	0	0	0	0	0
TOTAL		%	1 1.3%	47 61.8%	9 6.6%	24 30.3%	81 100.0%

Chi-square *p* value: 0.000. Kappa coefficient: 0.709.

**Table 2 medicina-58-00321-t002:** (**a**) The relationship between DWI results and surgery for tendons with partial tear; (**b**) the relationship between DWI results and surgery for tendons with tenosynovitis.

**(a)**										
**Tendon**				**Surgery Result**		**Total**	**Sensitivity**	**Specificity**	**Fisher’s *p* Value**	**Kappa Coefficient**
				**−**	**+**					
EHL	DWI-Partial Tear	−	*n*	1	0	1	100%	100%	1.000	1.000
			%	100.0%	0.0%	100.0%				
		+	*n*	0	1	1				
			%	0.0%	100.0%	100.0%				
	Total		*n*	1	1	2				
			%	50.0%	50.0%	100.0%				
EDL	DWI-Partial Tear	−	*n*	1	0	1	100%	100%	0.250	1.000
			%	100.0%	0.0%	100.0%				
		+	*n*	0	3	3				
			%	0.0%	100.0%	100.0%				
	Total		*n*	1	3	4				
			%	25.0%	75.0%	100.0%				
FDL	DWI-Partial Tear	−	*n*	1	0	1	100%	100%	1.000	0.400
			%	100.0%	0.0%	100.0%				
		+	*n*	1	1	2				
			%	50.0%	50.0%	100.0%				
	Total		*n*	2	1	3				
			%	66.7%	33.3%	100.0%				
FHL	DWI-Partial Tear	−	*n*	7	0	7	100%	88%	0.001 *	0.875
			%	100.0%	0.0%	100.0%				
		+	*n*	1	8	9				
			%	11.1%	88.9%	100.0%				
	Total		*n*	8	8	16				
			%	50.0%	50.0%	100.0%				
PT.	DWI-Partial Tear	−	*n*	7	0	7	100%	100%	0.000 *	1.000
			%	100.0%	0.0%	100.0%				
		+	*n*	0	8	8				
			%	0.0%	100.0%	100.0%				
	Total		*n*	7	8	15				
			%	46.7%	53.3%	100.0%				
**(b)**										
				**Surgery-Results**		**Total**	**Sensitivity**	**Specificity**	**Fisher’s *p* Value**	**Kappa Coefficient**
				**−**	**+**					
DWI-Tenosynovitis	+	*n*			1	1				
		%			100.0%	100.0%				
Total		*n*			1	1				
		%			100.0%	100.0%				
DWI-Tenosynovitis	−	*n*		1	0	1	100%	100%	0.125	1.000
		%		100.0%	0.0%	100.0%				
	+	*n*		0	7	7				
		%		0.0%	100.0%	100.0%				
Total		*n*		1	7	8				
		%		12.5%	87.5%	100.0%				
DWI-Tenosynovitis	−	*n*		2	0	2	100%	67%	0.107	0.714
		%		100.0%	0.0%	100.0%				
	+	*n*		1	5	6				
		%		16.7%	83.3%	100.0%				
Total		*n*		3	5	8				
		%		37.5%	62.5%	100.0%				
DWI-Tenosynovitis	−	*n*		1	0	1	100%	100%	0.333	1.000
		%		100.0%	0.0%	100.0%				
	+	*n*		0	2	2				
		%		0.0%	100.0%	100.0%				
Total		*n*		1	2	3				
		%		33.3%	66.7%	100.0%				
DWI-Tenosynovitis	−	*n*		1	0	1	100%	100%	0.091	1.000
		%		100.0%	0.0%	100.0%				
	+	*n*		0	10	10				
		%		0.0%	100.0%	100.0%				
Total		*n*		1	10	11				
		%		9.1%	90.9%	100.0%				

*: Statistically significant, *p* < 0.05.

**Table 3 medicina-58-00321-t003:** (**a**) The relationship between DWI and MRI results for all tendons with rupture; (**b**) the relationship between DWI and MRI results for all tendons with partial tear; (**c**) the relationship between DWI and MRI results for all tendons with tenosynovitis.

**(a)**									
**Rupture**			**MRI**		**Total**	**Sensitivity**	**Specificity**	**Fisher’s *p* Value**	**Kappa Coefficient**
			**−**	**+**					
DWI	−	*n*	3	0	3	100%	75%	0.048 *	0.769
		%	100.0%	0.0%	100.0%				
	+	*n*	1	5	6				
		%	16.7%	83.3%	100.0%				
Total		*n*	4	5	9				
		%	44.4%	55.6%	100.0%				
**(b)**									
**Partial Tear**			**MRI**		**Total**	**Sensitivity**	**Specificity**	**Fisher’s *p* Value**	**Kappa Coefficient**
			**−**	**+**					
DWI	−	*n*	16	1	17	92%	55%	0.006 *	0.362
		%	94.1%	5.9%	100.0%				
	+	*n*	13	11	23				
		%	54.2%	45.8%	100.0%				
Total		*n*	29	12	41				
		%	70.7%	29.3%	100.0%				
**(c)**									
**Tenosynovitis**			**MRI**		**Total**	**Sensitivity**	**Specificity**	**Fisher’s *p* Value**	**Kappa Coefficient**
			**−**	**+**					
DWI	−	*n*	4	1	5	97%	44%	0.017 *	0.459
		%	80.0%	20.0%	100.0%				
	+	*n*	5	21	26				
		%	19.2%	80.8%	100.0%				
Total		*n*	9	22	31				
		%	29.0%	71.0%	100.0%				

(Normal/healthy tendons were not evaluated). *: Statistically significant, *p* < 0.05.

## Data Availability

All available datas were reported for the research.

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
