# Peer review of "The Role of Diffusion Weighted MR Imaging in the Diagnosis of Tendon Injuries of the Ankle and Foot"

_medicina, 2022, doi:10.3390/medicina58020321_

Round 1

Reviewer 1 Report

Manuscript ID: medicina1558176

Manuscript title: The role of diffusion weighted MR imaging in the diagnosis of tendon injuries of the ankle and foot

Major comments

  1. Introduction (lines 53-59). This paragraph is unclear and may benefit from a revision. Consider being more explicit and objective when stating the hypothesis to be tested and the study aims. For instance, it is mentioned ‘correlated’ but there is no correlational analysis in the manuscript (Chi-Square tests do not measure correlation in the statistical sense).
  2. Methods. As a diagnostic accuracy study, the manuscript reporting would greatly improve if the specific guideline from Equator Network is followed (https://www.equator-network.org/reporting-guidelines/stard/).
  3. Methods (lines 61 and 78). Please provide additional information regarding the sample size and the number of tendon lesions for a better understanding of your unity of analysis. You mention 50 patients and 81 tendon injuries. I missed information about how many patients had at least one injury.
  4. Methods (lines 107-108). Tendon injuries were categorized into three levels (rupture or complete tear; partial tear; thickening), I assume with an additional category for ‘normal’ or ‘healthy’ tendon (confirmed later after seeing Table 1). How many ‘normal’ tendons were included? If only injured tendons were included, what reference value was used to test for diagnostic accuracy?
  5. Methods (lines 130-145). This text passage should be a new section of the manuscript (Statistical analysis). It also would benefit from a revision by an expert statistician. For instance: there are several kappa models, but it is unclear which one was used; the kappa coefficient is not a ‘variance analysis’ or ‘compatibility index’; Fisher’s exact test does not assess ‘correlation’.
  6. Methods (lines 139-140). Sensitivity and specificity are interesting properties of the new test, but positive/negative predictive values are much more informative for clinicians. Overall accuracy could be easily reported from the cross-tables. Also, it needs to be clearly stated what test is considered the gold standard and the new index for 2x2 cross-classifications.
  7. Methods (lines 142-145). The rationale for the ROC plot analysis is unclear. Why would you consider the confidence level of each rater as a new diagnostic test for tendon injury? Nonetheless, after checking Graphics 1 to 3 it is apparent that this was not the analysis you performed. Please double-check.
  8. Results. This section needs extensive revision due to the above-mentioned comments, in particular those related to the applied statistical methods and their interpretation.
  9. Results (Table 1). Just checking if ‘1. Observer’ report no ‘Normal’ tendon. As Table 1 total reads 76, I assume there is a missing line for this ‘normal’ outcome with 5 patients to include. Otherwise, to what categories do they belong?
  10. Results (Tables 3a to 3c). All tables sum to 81, the number of lesions reported. It does not seem methodologically correct to cross-tabulate only the injuries (+ results) and not consider the normal (- results). What is the rationale to analyze diagnostic probabilities only among the + test results?
  11. Lastly, the manuscript would benefit from professional English proofreading.

Author Response

Responses

1-Introduction part was briefly revised.Correlated item was also corrected.

2-Methods part wa sectioned wh regard to the specific guidelines from Equator Network

3- Methods (lines 61 and 78).were corrected  regarding the sample size and the number of tendon lesions, number of patients having at least one injury was also mentioned.

4-Normal/healthy tendons were not included in the research, reference values for diagnostic accuracy were mentioned. 

5-Statistical analysis was checked and all used tests and kappa coefficients were re-evaluated.

6- positive/negative predictive values, the gold standard test and the new index for 2x2 cross-classifications were mentioned.

7-ROC plot analysis was revised

8-Results part was extensively revised.

9-Table 1 was corrected.

10-Normal tendons were not included, diagnostic probabilities only among the + test results were defined.

11-Moderate English editing of the text was achievved.

All revisions were red-highlighted.

Reviewer 2 Report

The topic is one of importance given the high number of presentations to health services that are related to concerns on ankle sprains. Also, this is an interesting aim  to determine the diagnostic performance and utility of Diffusion Weighted MR Imaging (DWI) against the routine Magnetic Resonance Imaging (MRI) for the evaluation of 11patients with tendon injuries of ankle and foot. I think it would be a more clear study if the following parts were revised and supplemented.  These will be discussed below relative to the information of the manuscript.

General Comments:

Overall the manuscript is generally well written and is a topic of interest. However after reading it a number of times I am still left without key take-home points. I believe these points are in the results they just need to be developed.

Specific comments:

Abstract:

1) The authors state they will  to determine the diagnostic performance and utility of Diffusion Weighted MR Imaging (DWI) against the routine Magnetic Resonance Imaging (MRI) for the evaluation patients with tendon injuries of ankle and foot. This seems like too much of an over simplification of what was done. I do feel that it would be beneficial to explain what specifically you are looking at in relation to ankle sprains  (this also applies to the main body of the paper). Is it the development of MR-images literature. This needs to be made clearer throughout the paper. (Major Compulsory Revision)

Introduction

2) The first paragraph should have a sentence or two added that better outlines why this study is important related with magnetic resonance imaging (MRI) to determine the association between ankle tendon, ligament, and joint conditions and ATFL injuries https://pubmed.ncbi.nlm.nih.gov/33392013/ (Major Compulsory Revision)

The authors do a poor job on reviewing relevant literatura related with importance with ankle sprain is one of the most common injuries. Please revise the research of Kazzemi el al related with muscle activity  and Calvo el al and the importance of these muscles quantify the cross-sectional area (CSA) of the peroneus brevis, the peroneus longus, and connective tissue https://pubmed.ncbi.nlm.nih.gov/27793349/

3) In the last paragraph, the significance of the proposed word should be included highlighting why your work is important. what is the scientific contribution of this paper? it is not clear how this paper can make a significant contribution to the state of the art.  (Major Compulsory Revision).

In addition, author´s hypotheses should be included.

Operational definitions of intra- and inter- reliability should be included in introduction.

The authors do not explain why it is important to assess intra- and inter- rater reliability and how the evaluators or the time intervals could influence the measurements in this experiment. Please, explain it. What other studies say about this? Please, relate other studies regarding these tests.

4) Provide ethic committee approval number and organization. Major Compulsory Revision).

5) Who determined that the patients ankle MRI scan  met the inclusion criteria? A therapist? Include the information about it.

6) Where the experiments carried out? In a hospital? In an educational institution? Provide this information.

7) Test protocol to assess intra- and inter-rater reliability must be described. how many sessions were carried out during the experiment and how they were related to asses intra- and inter- rater reliability? Major Compulsory Revision).

how many trials were made during each session?

Which evaluator conducted each session?

The authors say that testers have different levels of MRI experience. A description of the testers must be provided.

8) Add a study flow chart for the readers. Major Compulsory Revision).

The measurements were carried out the same day? Which time interval between measurements was used to asses intra-rater reliability? was this continuous? on separate days? In the same day? and why was it carried out that way?

To calculate inter-rater, were the sessions carried out by the evaluators on the same day? The conditions during the experiment sessions for the two evaluators must be the same, how did they achieve this?

9) The score about reliability must be referenced.

10) The authors did not provide an interpretation scale for SEM. Please, provide it with an appropriate reference.

If the authors do not have a previous work that supports that their method is reliable with respect to other methods, then the reported measurements must also be obtained using another traditional method to assess inter-method reliability. (Major Compulsory Revision)

The Discussion section is a rehashing of the results. It does not appear that the authors include much interpretation of what the study findings mean for clinical practice or research. (Major Compulsory Revision)

FInally, the conclusión is weak and too long. (Major Compulsory Revision)

Author Response

Responses

1-What we looking for specifically in ankle sprains was briefly explained in the abstract section.

2- https://pubmed.ncbi.nlm.nih.gov/33392013/ Kazzemi et al and Calvo el al https://pubmed. ncbi.nlm. nih.gov/27793349/ brief informations about these articles were  added in the introduction part 

3- Scientific contribution of this paper was stated in the introduction part our hypotheses and definitions of intra- and inter-reliability were included  Assessment of  intra and inter-rater reliability with regard to other studies in the literature was performed

4- Ethic committee year-approval number and organization.was writtenin the text.

5- The information about the meeting of  inclusion criteria in the ankle MRI scan was described. 

6- The place of the experiments handled, was defined.

7- Test protocole for assessing intra- and inter-rater reliability was described, number of sessions and their route of assessing intra- and inter- rater reliability was explained.

Number of trials and evaluator conducting sessions were stated in the text.

MRI experience. of both observers  were described.

8- A study flow chart for the readers were included in the text.

Date of measurements, time interval between measurements assessment of intra-rater and inter-rater reliabilities were explained. Achievvement of  the evaluators was also briefly defined.

9- Scores about reliability were referenced

10- Interpretation scale for SEM was provided  with an appropriate reference.

Reliable method for reported measurements was regarded in the text.

Interpretation of the study for the clinical practice was briefly defined in the discussion part.

Conclusión part was strenghtened.

All revisions were red-highlighted.

Round 2

Reviewer 1 Report

Manuscript ID: medicina1558176.V2

Manuscript title: The role of diffusion weighted MR imaging in the diagnosis of tendon injuries of the ankle and foot

Thank you for the opportunity to assess the revised manuscript. Most of my previous comments were properly addressed, but a few remained. Please find below minor comments pending.

Minor comments

  1. Methods, Statistical Analysis section (lines 202-219). Kappa model is still not specified (Unweighted? Weighted?). The kappa coefficient is still labeled as a ‘variance analysis’ or ‘compatibility index’ (in tables). Fisher’s exact test is not a gold standard tests’ itself, but can test for associations between gold-standard and a new test.
  2. Methods (lines 142-145). The rationale for the ROC plot analysis was not provided. Why would you consider the confidence level of each rater as a new diagnostic test for tendon injury? Nonetheless, after checking Graphics 1 to 3 it is apparent that this was not the analysis you performed. Please double-check.
  3. Results (Table 1). For completeness, Table 1 must show for ‘1. Observer’ a row for the ‘normal’ outcome with 0 patients.
  4. Results (Tables 3a to 3c). What is the rationale to analyze diagnostic probabilities only among the + test results? This is not informative for clinicians as they will face both positive and negative test results.

Author Response

1-Kappa model is specified. The kappa coefficient was exactly described in tables. Gold standard test was re-established.

  1. The rationales for the ROC plot analysis were mentioned and the reason of chosen confidence level of each rater as a new diagnostic test for tendon injury was presented. Graphics 1 to 3 were re-checked.
  2. A row for the ‘normal’ outcome with 0 patients for Observer 1 was added to Table 1.
  1. The rationales for diagnostic probabilities of + test results were described for Tables 3a to 3c.

Reviewer 2 Report

The authors have clearly and adequately addressed almost all comments raised by the reviewers. Please update in the references section the references 5, 14, 17, 18, 21 22, 23, 24, 25.... using  the format adequate of this journal

Author Response

The references 5, 14, 17, 18, 21 22, 23, 24, 25 were corrected with regard to the  format of the journal.